# From the Structural and (Dys)Function of ATP Synthase to Deficiency in Age-Related Diseases

**DOI:** 10.3390/life12030401

**Published:** 2022-03-10

**Authors:** Caterina Garone, Andrea Pietra, Salvatore Nesci

**Affiliations:** 1Department of Medical and Surgical Sciences, Alma Mater Studiorum University of Bologna, 40137 Bologna, Italy; andrea.pietra@studio.unibo.it; 2Center for Applied Biomedical Research, Alma Mater Studiorum University of Bologna, 40137 Bologna, Italy; 3UOC Neuropsichiatria dell’età Pediatrica, IRCCS Istituto delle Scienze Neurologiche di Bologna, 40137 Bologna, Italy; 4UO Genetica Medica, IRCCS Azienda Ospedaliero-Universitaria di Bologna, 40137 Bologna, Italy; 5Department of Veterinary Medical Sciences, Alma Mater Studiorum University of Bologna, 40064 Ozzano Emilia, Italy

**Keywords:** mitochondria, ATP synthase, cell death, neurodegenerative diseases

## Abstract

The ATP synthase is a mitochondrial inner membrane complex whose function is essential for cell bioenergy, being responsible for the conversion of ADP into ATP and playing a role in mitochondrial *cristae* morphology organization. The enzyme is composed of 18 protein subunits, 16 nuclear DNA (nDNA) encoded and two mitochondrial DNA (mtDNA) encoded, organized in two domains, F_O_ and F_1_. Pathogenetic variants in genes encoding structural subunits or assembly factors are responsible for fatal human diseases. Emerging evidence also underlines the role of ATP-synthase in neurodegenerative diseases as Parkinson’s, Alzheimer’s, and motor neuron diseases such as Amyotrophic Lateral Sclerosis. Post-translational modification, epigenetic modulation of ATP gene expression and protein level, and the mechanism of mitochondrial transition pore have been deemed responsible for neuronal cell death in vivo and in vitro models for neurodegenerative diseases. In this review, we will explore ATP synthase assembly and function in physiological and pathological conditions by referring to the recent cryo-EM studies and by exploring human disease models.

## 1. Introduction

Cell survival relies on energy production in the form of ATP (adenosine tri-phosphate) molecules, the universal energy currency for all living organisms. The mitochondria, known as the cellular powerhouse, are the energy-producing organelles of the cell [1]. Unique characteristics are the presence of DNA (mtDNA) exclusively inherited from the mother. The dual control of both nuclear (nDNA) and mtDNA is responsible for encoding around 1500 proteins, playing role in mitochondrial function and maintenance. ATP synthase (EC 3.6.1.34), also known as complex V, is composed of 18 protein subunits, 16 nDNA encoded and two mtDNA encoded. The main function is the conversion of ADP into ATP, thanks to the proton gradient generated by complexes I to IV and activating the rotor mechanism. Being essential in aerobic cells, the basic structure and catalytic mechanisms of ATP synthase are highly conserved across species, from bacteria to eukaryotes [2,3]. Moreover, the enzyme is involved in the morphology of mitochondria by contributing to the generation of inner membrane *cristae*, an event that evolves to include membrane “supernumerary” subunits [4]. The ATP synthase is a key participant in the cell bioenergetic machinery [5] by highlighting the prominent feature of the “enzyme of life”. Otherwise, the mitochondrial dysfunction may arise from the molecular switch of the ATP synthase function that occurs with the “supernumerary” subunits modification by stimulating different forms of regulated cell death [6]. The physio(patho)logical phenomenon might include the Ca^2+^-dependent permeability transition of the mitochondrial inner membrane to ions and solutes with molecular mass up to about 1.5 KDa [7,8,9]. Evidence highlights the role of this fascinating enzyme complex as a key molecular and enzymatic switch between cell life and death and increases its attractiveness as a pharmacological target and drug design [10].

However, mitochondria can take part in both the development and cell death process, in which features are dependent on the phenomenon that forms a high-conductance channel known as mitochondrial permeability transition pore (mPTP) [11].

The increase in the older population in modern society has become a problem in terms of socioeconomic burdens due to higher incidences of age-related neurodegenerative diseases. Therefore, the development of new strategies targeting these pathological features is a timely topic. Poor energy homeostasis linked to mitochondrial dysfunction springs from an impaired or defective energy transduction system, which constitutes the main biochemical damage in a variety of genetic and neuropsychiatric diseases [12].

In several neurodegenerative diseases, protein aggregation and mitochondrial dysfunction are two pathogenic processes responsible for the onset of age-related diseases. Indeed, mitochondrial structural and functional defects are attributable to mitochondrial dynamics impairment leading to neurodegeneration and ageing, Alzheimer’s disease (AD), Parkinson’s disease (PD), amyotrophic lateral sclerosis (ALS), Huntington’s disease (HD). The inhibition of excessive fission reduces the cell death pathway triggered by mitochondrial dysfunction [13]. Therefore, the morphology of mitochondria is intimately involved in ageing by revealing profound age-dependent changes in membrane architecture. The ATP synthase is involved in generating mitochondrial *cristae* morphology [4]. ATP synthase dimers are arranged in long rows on the tip of *cristae* together with the MICOS (mitochondrial contact site and *cristae* organizing system) complex at *cristae junctions* and cooperate by opposing effects on membrane curvature to the *cristae* morphology [14]. The membrane remodelling triggered by the disassembly of the supramolecular organization of ATP synthase resembles the inner-membrane vesiculation of old mitochondria. In all likelihood, the *cristae* shorten and collapse with the ATP synthase dimers dissociation that reduces the convex membrane curvature in the inner mitochondrial membrane (IMM).

Age-related decline in bioenergetics, redox homeostasis, and mitochondrial calcium capacity contribute to accelerating pathogenesis. Indeed, the failure of the organisational power of energy flow in cells is the basis of the biological complexity of diseases in ageing [15].

## 2. The Structure and Function of the ATP Synthase

The ATP synthase is a ubiquitous oligomeric complex placed in the energy-transducing membranes of mitochondria, chloroplasts, and bacteria [16]. The transmembrane protonmotive force (Δ*p*) is employed by ATP synthase as a source of energy. The Δ*p* drives the mechanical rotary mechanism by the membrane-embedded portion of the enzyme (F_O_) coupled to the chemomechanical synthesis of ATP from ADP and Pi on the hydrophilic F_1_ portion (Figure 1). Moreover, the Δ*p* generated by substrate oxidation during the mitochondrial respiration is dissipated by F_O_ with torque generation of the rotor permitting the cooperative ATP binding change mechanism of catalytic sites during ATP production [17]. Conversely, during the ATP hydrolysis, the enzyme works in reverse as an H^+^ pump and re-energises the IMM. This bifunctional energy transmission mechanism of ATP synthase is unique in biological systems [18]. The membrane-bound rotor consists of a *c*-ring of eight identical helical hairpin structures in close contact with the *a* subunit. The *c*-ring stoichiometry is species-dependent and tightly related to the mitochondrial bioenergetic cost [2,3]. Each *c* subunit has an H^+^-binding site defined by a conserved acidic side chain (Glu58 in mammalian) on the C-terminal helix in the middle of IMM that is dicyclohexylcarbodiimide-sensitive and essential in ATP synthesis or hydrolysis. The H^+^ translocation takes place between two aqueous half-channels in *a* subunit [19], in which asymmetric arrangement dictates the two opposite rotation directions in the ATP synthesis and hydrolysis [20]. The rotation of the *c*-ring is transmitted directly to the F_1_ domain by the asymmetrical central stalk (composed of γ, δ, and ε subunit) attached by the “foot” to the *c*-ring. The γ subunit of the central stalk penetrates along the central axis of the F_1_ domain that contains three non-catalytic α- and three catalytic β-subunits alternate in a hexameric (αβ)_3_ structure around the central stalk (Figure 1).

The catalytic sites are placed on the β subunits at the interface with the α subunits. Each β subunit catalyzes the reaction of synthesis or hydrolysis driven by Δ*p* or ATP phosphorylation potential, respectively. During the synthesis, the rotation of the rotor view from the matrix side is in anticlockwise mode. Conversely, the clockwise rotation of the rotor supports ATP hydrolysis. Each β subunit transforms an ATP molecule in a complete rotation (360°). Therefore, the ATP synthase produces or hydrolyzes three ATP molecules per cycle. This coordinate mechanism is the molecular event of a “*Splendid Molecular Machine*” supported by a cooperative binding change mechanism of the (αβ)_3_ structure [21]. Indeed, the catalytic sites can assume three distinct conformations characterized by a different affinity for adenine nucleotide. The β_TP_, β_DP_, and β_E_ have three nucleotide bound-states: the first has ATP or ATP analogues in the catalytic site, the second has ATP or ADP, and the third has no bound nucleotide. The affinity for adenine nucleotide decreases in the absence of ATP and permits the “closed” conformation with the β_TP_, β_DP_ states or “open” state with β_E_. The enzyme kinetics of the catalytic sites require nucleotide coordination with the essential cofactor Mg^2+^ [22], which contributes to the catalytic site asymmetry, producing the different affinities for ATP [23]. On the contrary, all of the α subunits that are bound to the ATP molecule coordinate to the Mg^2+^ cofactor and do not undergo chemical transformation. The MgATP bound to the non-catalytic sites has an important role in permitting the detachment of ADP from the β sites. In particular, the ATP hydrolysis forms an β_E_ state loaded with ADP in a “half-closed” conformation that could inhibit the ATP synthase in the absence of MgATP into the non-catalytic sites [24].

Recently, Pinke et al. have revealed a cryo-EM structure of ATP synthase with the natural cofactor Mg^2+^ substituted in the catalytic sites by the Ca^2+^ [25]. The ATP synthase activated by Ca^2+^ as a cofactor sustains only the monofunctional activity of ATP hydrolysis, whereas ATP synthesis is not permitted. In inside-out submitochondrial particles, the hydrolysis of ATP in the presence of Ca^2+^ determines the acridine fluorescence unquenching [26]. Conversely, the ATPase activity is modulated by the Δ*p* [27]. For these opposite results, the Ca^2+^-dependent ATP hydrolysis coupled to H^+^ pumping activity is debated, even if the unidirectional rotational catalysis is retained [28]. However, the pathophysiological role of Ca^2+^-supported ATP hydrolysis should not be investigated in the activity of ATP synthase (un)coupled to H^+^ translocation, but in the induction of conformational changes that lead to the formation of mPTP [29].

Importantly, a static structure peripheral to the central rotor, which spans the entire ATP synthase length, acts as a stator to prevent the (αβ)_3_ rotation torque by the central stalk. The peripheral stalk (PS) ensures the energy transduction mechanism of the enzyme with the hydrophilic F_1_ domain enzyme activity coupled to H^+^ translocation in the hydrophobic F_O_ domain [22,30]. The PS is composed of hydrophilic and hydrophobic subunits. The formers are the oligomycin sensitivity-conferring protein (OSCP), F6, the soluble portion of the *b*, and *d* subunit bound to (αβ)_3_ hexameric structure of the F_1_ domain, whereas the hydrophobic portion of *b* and A6L subunits form the embedded portion in the IMM of the PS (Figure 1). However, prevention of idle rotation of the globular catalytic structure of the F_1_ domain with the rotor is allowed by exploiting the elastic feature of the PS. The interdomain hinge movement of the OSCP can facilitate the flexible coupling of F_1_ and F_O_ [31].

Most of the membrane subunits of the enzyme are encoded by nDNA. Conversely, the membrane subunits identified as *a*, A6L in the mammalian ATP synthase are encoded by the mtDNA. In yeast, the *c* subunit is also encoded by the mtDNA. The latter, expressed by nDNA in mammals, has three different isoforms. Notably, different supernumerary subunits structurally contribute to the mitochondrial ATP synthase dimerization and IMM curvature (Figure 2) [32,33]. In the mammalian ATP synthase, these subunits are identified as 6.8-kDa proteolipid (6.8 PL), diabetes-associated protein in insulin-sensitive tissues (DAPIT), A6L, *e*, *f*, and *g* and there is one copy of each protein per ATP synthase monomer. The DAPIT, A6L, and *e* subunits have a single predicted transmembrane α-helix, but the latter has a portion of the helix that protrudes into the intracristae space [25].

Two monomers of ATP synthase are joined together, and some subunit–subunit interactions are involved in IMM bending. The convex angle of 112° at the tip of the *cristae* (Figure 2) arises from a reminiscent BAR-like domain [34] that bends the lipid bilayer to the highly curved apex of the *cristae*. The structural composition of the subunits includes the single helix (H) of the e subunit with its transmembrane portion, the third α-helix of the three α-helices of the *g* subunit (H3*g*), and the H2 and H3 of the *b* subunit. The topology of the Hs distribution in the BAR-like domain is responsible for a 61° angle of IMM (Figure 2). Three distinct contact sites take place between two monomers during the ATP synthase dimerization: two of them laterally to the dimer, respectively in the inner layer of the IMM, facing the matrix, and in the outer layer facing the intracristae space; whereas a third site is located in the middle of the supramolecular structure. The subunits involved in the interaction are the *f* and the 6.8 PL subunit of neighbour monomers of the same dimer at the matrix side. The contact site establishes between *e,* and the 6.8 PL of adjacent monomers from the dimer are placed at the intracristae space side. The contact sites in the dimeric core of ATP synthases are arranged between opposite subunits of monomers at the matrix side between *f* subunits within the IMM, in the middle of the membrane between *a* subunits, and at the intracristae space between *e* subunits (Figure 2).

The ATP synthase dimerization together with protein-lipid interaction is fundamental for *cristae* shaping [4,22]. While ATP monomers are mobile and localized at the inner boundary membranes, the dimers are able to organize themselves in rows giving the protein a spontaneous curvature essential for localization at the rim of the high-curvature edges of the inner mitochondrial membrane. Deletion or downregulation of subunits involved in monomers interactions causes absent or balloon-shaped *cristae* [35,36]. Moreover, while ATP monomers have hydrolytic activity, dimers are responsible for ATP-synthase function [37]. The mechanical adaptation of the membranes into *cristae* is favourite by the interaction between cardiolipin and ATP dimers. Cardiolipin creates microenvironments promoting ATP synthase dimerization and organization in ribbon-like structures and it seems to play the role of H^+^-trap that optimize the ATP synthase activity. Interestingly, positive correlations have been found between the number of *cristae*-junctions, the width of *cristae* and ATP molecules while a negative correlation between ATP hydrolysis activity and mitochondrial morphology [38].

The role of ATP synthase in *cristae* shaping has also been recently linked to the purifying selection of mtDNA. Studies in *S. Cerevisiae* have demonstrated that mtDNA segregation requires a continuous mitochondrial network and ATP synthase dimers stand in close proximity of mtDNA creating a compartment where different mtDNA molecules are not mixed. This “compartmentalization” is responsible for the negative selection of mutant mtDNA molecules that cannot be complemented by wild-type mtDNA molecules and create a dysfunctional bioenergetic environment that leads to mtDNA purging [39].

## 3. The Hidden Face of the ATP Synthase

Even if the overall architecture, organization and mechanistic principles of ATP synthase have been well established in the last decade, other roles of ATP synthase in mitochondrial biology are less well understood. Indeed, the ATP synthase function is not only confined to ATP production but also, as recently emerged, when activated by Ca^2+^ [27,29,40,41], in the formation of the lethal mega-channel in the IMM known as the mPTP [42]. The mPTP would coincide with the ATP synthase [25]. When in the catalytic site the natural cofactor Mg^2+^ is replaced by Ca^2+^, the enzyme would disassemble, forming the mPTP in the F_O_ domain. Since the Ca^2+^-activated ATP synthase responds differently to various modulators with respect to the enzyme activated by natural cofactor Mg^2+^, biochemical strategies targeting the enzyme may potentially rule the mPTP. On one side to prevent its opening and the associated deadly events, and on the other, to trigger its formation to selectively counteract cell proliferation. This arcane phenomenon raised the research interests in mPTP modulators [43,44,45,46,47,48,49], which can represent innovative drugs to counteract a variety of mPTP-related human diseases [10,50]. Kinetic analyses in vitro on the ATP synthase inhibition mechanisms and the interaction between different inhibitors, under different assay conditions, have been used as a biochemical strategy to identify and evaluate the effect of post-translational modifications of amino acids (e.g., Cyst, Tyr, His, and Arg) on the molecular mechanisms of energy transduction in mitochondria [43,48,51,52,53,54].

Different hypotheses have been proposed to explain the role of ATP synthase as a pore-forming component of the mPTP. The opening of a large conductance channel could be triggered by the *c*-ring of ATP synthase [55,56] or it forms in ATP synthase dimers at the interface between monomers [57,58]. However, the recent cryo-EM structure of ATP synthase with the Ca^2+^ bound to catalytic sites of the enzyme [25] supports the “bent-pull” model [41,59]. So, the mPTP opening arises from different conformations of the Ca^2+^-activated ATP synthase and the pore forms from the *c*-ring. The Ca^2+^ has a higher atomic radius than the natural cofactor Mg^2+^ by inducing an alteration of the F_1_ domain. The structural modification of the enzyme catalytic sites is transmitted by the PS to the F_O_ domain. The conformational changes generate a force that pulls the lipid plug out of the *c*-ring. In all likelihood, the lyso-phosphatidylserine (L-PS) on the intracristae space is pushed out of the hole of the *c*-ring by the movement induced by the *e* subunit. Consistently, the water molecules fill the inside of the *c*-ring and destabilise the phosphatidylserine plug on the matrix side. Accordingly, the F_1_ detaches from the F_O_ and the pore opens [25,41] (Figure 3).

The Ca^2+^ signalling propagation pathway of the Ca^2+^-activated ATP synthase that triggers the mPTP is supported by the unit structure of *b*, *g*, and *e* subunit. This structure, known as “hook apparatus”, performs the mechanism of mPTP formation. Indeed, the conformational changes of ATP hydrolysis supported by Ca^2+^ as a cofactor in the F_1_ domain would generate the force transmitted along with the *b* subunit from the soluble hydrophilic subunits to membrane subunits. In the F_O_ domain, the H2*b*, H3*g*, and H*e* helixes are joined together by salt bridges and are assembled in a triple transmembrane helix bundle (TTMHB) (Figure 4). Moreover, in the TTMHB, the *e* and *g* subunits are tightly bound by the GxxxG motifs [25]. An L-PS is attached to the end of the α-helix of *e* subunit, which fills the hole of the *c*-ring and acts as a lubricating lipid plug allowing rotation of the rotor. The pushing forces on the long helix of the *b* subunit, which are generated by conformational changes in the F_1_ domain loaded with Ca^2+^, pull the L-PS plug out of the *c*-ring by the hook (Figure 4). Moreover, the modification of the membrane subunits arrangement in the IMM, when the mPTP phenomenon is triggered, changes the subunit–subunit interaction responsible for dimer formation and membrane curvature at the ridge of the *cristae*.

## 4. The ATP Synthase Deficiency

Mitochondrial integrity and function are crucial for the proliferation, differentiation, and maintenance of neural stem cells (NSCs) in health and disease. During embryonic and adult neurogenesis, NSCs undergo a metabolic switch from glycolytic to oxidative phosphorylation (OXPHOS), with a rise in mtDNA content, changes in mitochondria shape and size, and a physiological augmentation of mitochondrial reactive oxygen species (ROS) which together drive NSCs to proliferate and differentiate. Defects in nDNA or mtDNA OXPHOS subunits can affect the mechanism responsible for controlling NCS fate, and cause neurodevelopmental disorders, brain malformation, and impaired regenerative capacity in adult stem cells. Moreover, whenever a metabolic requirement goes beyond a sustainable level of metabolic flexibility, mitochondrial dysfunction challenges cellular resilience and brain function, and contributes to neurodegeneration [60].

ATP synthase plays a central role in maintaining an adequate bioenergetic metabolism of neural cells other than controlling the mitochondrial dynamic and oxidative ROS production. Therefore, ATP synthase deficiency can be responsible for inherited mitochondrial encephalopathy and contributes to neurodevelopmental and neurodegenerative disorders. Dysfunction may result from structural or enzymatic defects due to aberrant mechanisms of synthesis, assembly, biogenesis and regulation of the ATP synthase complexes [61]. Direct “hits” are genetic variants in nDNA or mDNA encoded subunits causing maternal or mendelian inherited mitochondrial disorders. Contrariwise, indirect hits are defects in nDNA encoded assembly factors or regulatory and interacting proteins. Although the underlying disease mechanism is different, they all culminate with reduced catalytic activity and profound alteration of mitochondrial morphology with onion-like remodeling of *cristae* shape.

MT-ATP6 defect is the most common cause of ATP synthase deficiency. This is associated with maternally inherited mitochondrial encephalopathy with a predominant tissue-specificity for the cerebellum and basal ganglia and frequent multisystemic involvement of muscle, heart, peripheral nervous system, hearing and eye. Clinical features include ataxia, motor and language developmental delay, deafness, retinitis pigmentosa, hypertrophic cardiomyopathy. The first and most common pathogenetic variants, m.8993T>G (p.Leu156Arg) and m.8993T>C (p.Leu156Pro), were described in families presenting the spectrum of Maternal Inherited Leigh syndrome (MILS) and neurogenic muscle weakness, ataxia, retinitis pigmentosa (NARP) [62,63]. The biochemical defect is more pronounced in the m.8993T>G, introducing a positively charged amino acid (Arg) in a very highly conserved position (Leu156), consequently affecting the *c*-ring rotation and reducing the proton flux through F_O_ [64]. More than 70 variants have been described in the literature, 10 of them with confirmed pathogenicity in several independent research studies (www.mitomap.org) [65]. Clinical, neurophysiological and neuroimaging features depict an additional three major syndromes: Familiar bilateral striatal necrosis characterized by bilateral symmetrical degeneration of the basal ganglia, predominantly of the caudate and putamen nucleus; [66,67]; Leber Hereditary Optic Neuropathy (LHON) characterized by acute or subacute central vision loss leading to central scotoma and blindness [68]; Charcot-Marie-Tooth Neuropathy described in patients with pes cavus, subtle gait unsteadiness, weakness, atrophy and paraesthesia and sensory axial neuropathy [69,70,71].

Pathogenetic variants of MT-ATP8 are rarer causes of ATP synthase deficiency. Only eight patients have been described presenting with hypertrophic cardiomyopathies, cerebellar ataxia and psychomotor delay the majority of them [70,72]. Additional signs and symptoms were deafness, seizures, spastic paraplegia, neuropathy. Interestingly, Fehli et al. reported a patient with psychomotor delay, global and axial hypotony and seizures, and bilateral abnormalities of signal intensity in the bilateral lenticular nucleus, dentate nucleus and periventricular, carrying two homoplasmic variants: m.8392C>T (Pro136Ser) in MT-ATP6 gene and m.8527A>G at the junction MT-ATP6/MT-ATP8 [73].

Autosomal recessive ATP synthase deficiency has been associated with “direct hits” due to pathogenetic variants in four nuclear-encoded subunits: ATP5A1 encoding for the α-subunit in F_1_ subcomplex; ATP5D and ATP5E, respectively encoding for δ and ε subunit, form the central stalk with γ subunit; USMG5 encoding for a protein involved in the dimerization of ATP synthase.

Although they all present with encephalopathy [74,75,76,77,78,79], a very early onset (birth-day 1 of life) with rapid progression to fatal *exitus* characterized ATP5A1 defect [74,75]. Episodes of metabolic decompensation, consciousness deterioration, psychomotor regression with partial resolution of clinical, and neuroimaging features were the striking characteristics of the encephalopathy presented by patients carrying pathogenetic variants in ATP5D [76] and USMG5 [79].

Only two “indirect hits” due to defects in assembly factor (TMEM70, ATPAF2) cause ATP synthase deficiency. ATPAF2 is a nuclear gene encoding a protein that binds single α subunits and allows their bond with β subunits to avoid their aggregation in the matrix. A homozygous mutation c.280T>A (p.Trp94Arg) was found in a girl with dysmorphological features (wide mouth, prominent nasal bridge, micrognathia, rocker-bottom feet and flexion contractures associated with camptodactyly), hepatomegaly, renal hypoplasia, abnormal urinary excretion of lactate, fumarate, methylglutamic acid and amino acids, rapidly progressing cortico-subcortical and basal ganglia atrophy with dysgenesis of corpus callosum and hypomyelination. The girl died at 14 months of age for recurrent infections [80].

Homozygous variants in the TMEM70 gene typically cause hypertrophic cardiomyopathy, arrhythmia, encephalopathy, dysmorphological features, hypotonia, ataxia, difficulty to thrive, psychomotor delay, severe lactic acidosis and 3-MGA [81]. Additional signs and symptoms are microcephaly [82], cataract [83], ptosis [84], myoclonic episodes and an increase of Ala, Gln/Glu ratio and ammonium in blood [85] MRI brain showed mild cortical atrophy in some patients [83]. A metabolic crisis may occur at any disease stage and can lead to a fatal outcome [86].

Clinical signs and symptoms of ATP synthase deficiency are summarized in Table 1.

There is also increasing evidence of a common mitochondrial energetic dysfunction in neurodevelopmental disorders. Several genetic disorders whose defects were not primarily localized to mitochondria have been found to be linked to mitochondrial dysfunction [99]. Fragile X syndrome, a neurodevelopmental disorder characterized by moderate to severe mental retardation, macroorchidism, and distinct facial features, is caused in the majority of cases by an unstable expansion of a CGG repeat in the FMR1 gene and abnormal methylation, which results in suppression of FMR1 transcription and decreased protein levels in the brain [100]. Indeed, in vivo and in vitro models for Fragile X syndrome have demonstrated that FMRP protein regulates closure of the ATP-synthase *c* subunit leak. Neural and astrocytes cells from Fmr1−/y mice had inefficient respiration and an abnormal “proton leak”, which lead to ineffective OXPHOS [101,102]. Reduced ATP level and mitochondrial respiratory chain defect of complexes III and ATP synthase have been also found in brain tissue of cdkl5 null mice, a disease model for Rett syndrome [103,104].

## 5. Neurodegeneration

The uncoupling of the ATP synthase and hydrolysis processes, dissipation of mitochondrial inner membrane potential, elevated levels of ROS, low matrix content of ATP in combination with other cellular malfunctions trigger the opening of the mPTP in the IMM, one of the mechanisms responsible for cell death [105].

The opening of the mPTP is activated by a high concentration of Ca^2+^ in the mitochondrial matrix, inorganic phosphate, cyclophilin D (CypD) and ROS and is linked with the mitochondrial permeability transition (mPT), a phenomenon characterized by the sudden change in the IMM permeability to solutes with a size of <1.5 kDa [7,9]. When mPT occurs, the mechanism of the OXPHOS is uncoupled by the loss of the membrane potential, and the mitochondria undergo swelling and rupture of the outer mitochondrial membrane for osmotic dysregulation with consequent cell death. Other than Ca^2+^ accumulation inside mitochondria, mPT requires a protein forming the core channel part of mPTP. Several proteins such as VIDAC, ANT and CypD have been considered a potential forming channel, but robust results are for ATP synthase dimers or its *c*-ring, the major component of the membrane F_O_ complex of the enzyme, as the core channel of mPTP [55,106,107]. Indeed, high Ca^2+^ concentration inhibits the organization of *c* subunit in fibrils and induces the formation of cross-β oligomers, able to form small channels in model lipid bilayers [55,106,108,109].

The mPT phenomenon together with post-translational modifications and epigenetic modulation of ATP synthase subunits are the mechanism responsible for mitochondrial bioenergetics and dynamics dysfunctions. These are found in brain samples and in vitro and in vitro models of neurogenerative disorders. Amilodogenic proteins such Aβ-amiloid and α-synuclein can trigger mPT by direct or indirect activation of ATP synthase contributing to the mechanism of ageing-related disorders.

Parkinson’s disease (PD): degeneration of adrenergic neurons in substantia nigra pars compacta and striatum regions cause PD, clinically characterized by motor deficits such as bradykinesia, resting tremor, rigidity, and postural instability [110]. Affected neurons show intracellular deposition of Lewy bodies mostly formed by α-synuclein. Studies in rodents and biochemical and histological analysis of patient brain tissues demonstrated mitochondrial OXPHOS deficiencies as well as mitophagy and mtDNA alterations, suggesting the role of mitochondrial dysfunction in the disease pathogenesis. iPSC-derived mutant PARK2 neurons exhibited increased oxidative stress and aberrant activation of the NRF2 pathway with impaired mitochondrial function and α-synuclein accumulation [111]. Similarly, iPSC-derived mutant PARK2 dopaminergic neurons, carrying a GFP expression cassette at the TH locus, showed small and less functional neurons with a decline in mitochondrial membrane potential. In all likelihood, a mitochondrial-dependent mechanism is involved in dopaminergic neuron susceptibility to PD [112]. A potential link between α-synuclein and impairments in mitochondrial dynamics and bioenergetics is suggested by the direct interaction with the OXPHOS components, as demonstrated in proteomic studies [113,114]. α-synuclein aggregates can localise to the mitochondrial membrane and come into close proximity with a number of key mitochondrial proteins such as Complex I and ATP synthase [115,116]. Studies in isolated mitochondria, rodent neurons, and human iPSC-derived neurons demonstrated that under physiological conditions, monomeric α-synuclein interacts with ATP synthase α subunit. Consistently, ATP synthase activity and mitochondrial function increase while oligomers interaction with the ATP synthase β subunit induce selective oxidation and mitochondrial lipid peroxidation. The effects on the catalytic activity of ATP synthase potentially induce its mPTP-forming activity [117]. In inherited PD, due to PARK7 gene defect, the defective protein DJ-1 exerts a negative effect on ATP production, consequently causing mitochondrial dysfunction. Indeed, in PARK7 mouse model it has been demonstrated that the interaction with the wild-type form of DJ-1 decreased the mitochondrial uncoupling and enhanced ATP production, while defective DJ-1 increased mitochondrial uncoupling and depolarized neuronal mitochondria. The abnormalities of the ATP synthase in DJ-1^−/−^ animals consisted of reduced ATP levels and enzymatic function linked to impairment of neurite extension in isolated dopaminergic neurons. These results suggest that normal ATP synthase function is critical for dopaminergic neuron growth [118].

Alzheimer’s disease (AD): AD is one of the most common neurodegenerative disorders. AD clinically shows impairment of memory and executive functions with potential and variable involvement of other domains such as language, visual, or executive functions, are also possible [119]. Pathogenesis is characterized by extracellular β-amyloid plaques (Aβ) and intracellular hyperphosphorylated tau tangles [120] that cause synaptic dysfunction and neural cell death [105]. Multiple mitochondrial mechanisms may contribute to sporadic AD development, including reduced glucose and oxygen metabolism, altered mitochondrial morphology (*e.g.,* accumulation of osmophilic material, lipofuscin vacuoles, cristae disruption), defective complex IV function, and reduction in mtDNA content [119].

Aβ can enter into the mitochondrial matrix through the Translocase of the Outer Mitochondrial Membrane 40 (TOM40) import channel and the Translocase of the Inner Mitochondrial Membrane 23 (TIM23) import channel, which directly interact with the ATP synthase or indirectly influence its function and stability by interacting with CypD. Consequently, the amyloidogenic protein triggers the mPT contributing to neurodegeneration [121,122].

The demonstrated cross-link between the CypD and the extrinsic part of the PS at the top of the ATP synthase involved the OSCP, *b*, and *d* subunit [123]. In particular, the OSCP interdomain hinge movement might be blocked by CypD by reducing the flexibility between F_1_ and F_O_, which prevents the PTP opening. The molecular mechanism that connects the ATP synthase dysfunction and AD is linked to the physical interaction of Aβ with OSCP. The levels of OSCP subunits in mitochondria are inversely correlated with Aβ-OSCP interactions leading to ATP synthase deregulation in the development of AD mitochondrial defects [122]. The OSCP aberrations, which the Aβ toxicity induces, substantially sensitize the formation of mPTP.

ATP synthase catalytic activity is reduced in AD models by post-translational modification such as glycosylation or nitrosylation. Either an excess of nitrated α subunit or reduced glycosylation of the O-linked β-N-acetylglucosamine (O-GlcNAcylated) on the Thr432 residue of α subunit have been associated with reduced ATP synthase activity respectively in the hippocampus of AD patients, Tg AD mice and in Aβ-treated mammalian cell cultures–resulting in reduced ATP levels [124,125]. Moreover, 4-hydroxy-2-nonenal (4-HNE) was demonstrated to modify the α subunit of ATP synthase and decrease ATP hydrolysis in the hippocampal tissue and in the entorhinal cortex of early-stage AD individuals with mild cognitive impairment. The results suggest that oxidative stress precedes the presence of Aβ in the affected tissue and a 30%-decreased ATP synthase [126].

Gene expression and protein level analysis of ATP synthase subunits demonstrated an aberrant pattern characterized by a non-uniform increase or decrease in the nuclear and or mitochondrial encoded subunits in patients’ brain samples and in vitro and in vivo models. However, further studies are needed to confirm and clarify the mechanism responsible for those variabilities [61].

## 6. Conclusions

To sum up, the functional and bioarchitecture at the molecular and supramolecular structures of ATP synthase reveal the main role in physio(patho)logical conditions. Recent studies have shed light on the mechanisms linking ATP synthase dysfunction and neurodegeneration thereby identifying potentially druggable targets. This will enable the design of new therapies for ageing-relating disorders that will be exploited in further studies in vitro and in vivo models.

## Figures and Tables

**Figure 1 life-12-00401-f001:**
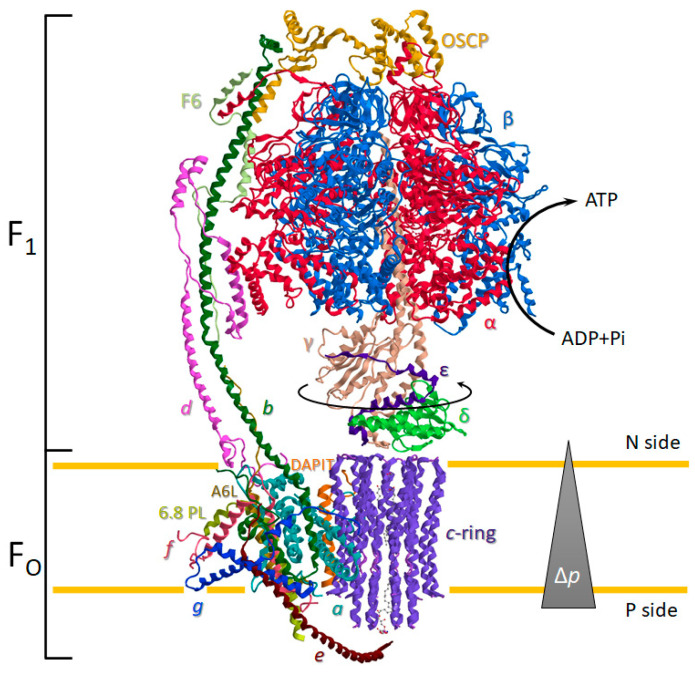
The overall structure of ATP synthase monomers in mammalian mitochondria. Enzyme subunits are drawn as ribbon representations obtained from modified PDB ID codes: 6TT7 [25]. The ATP synthesis is sustained by protonmotive force (Δ*p*). The differently coloured letters identify the subunits, drawn in the same colour as the letter. The matrix is the negative side (N side), whereas the intracristae space is the positive side (P side).

**Figure 2 life-12-00401-f002:**
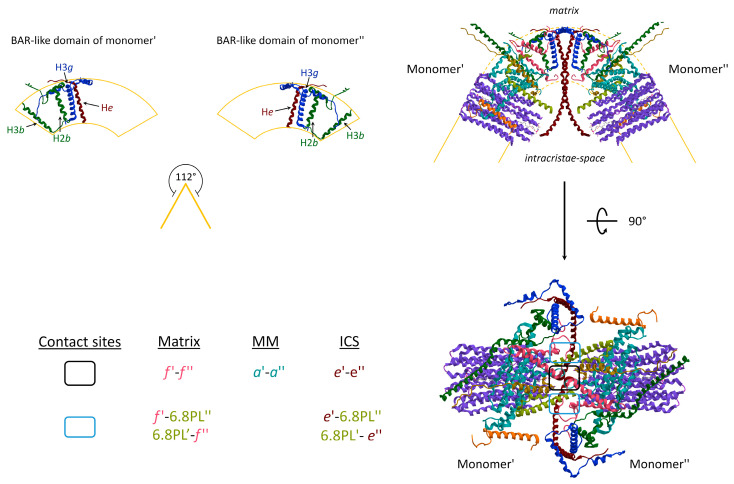
Membrane embedded-domains structure of ATP synthases (obtained from modified PDB ID code: 6TT7). The dimer assembly of monomers in the IMM causes the positive membrane curvature at the *cristae* tip (**upper panel**). The transmembrane α-helix (H) of subunits that form the BAR-like domain of each monomer are indicated. The angular association of monomers induces the strong curvature of the IMM with indicated angles of 112°. The contacts at the dimer interface are identified inside the boxes (**bottom panel**). The letter colours are the same as those of the subunits to which the structures belong. MM, middle of the membrane; ICS, intracristae space.

**Figure 3 life-12-00401-f003:**
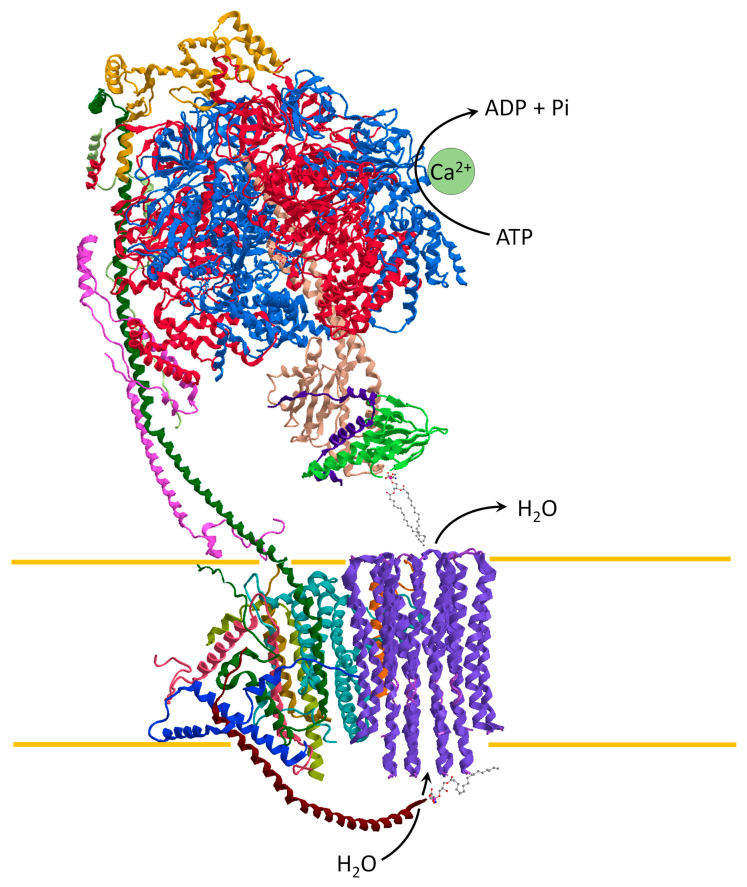
ATP synthase representation, obtained from modified PDB ID code: 6TT7, in the conformation of the mPTP opening. The Ca^2+^ bound to the catalytic sites drives the ATP hydrolysis by triggering the structural change which forms the mPTP. The pore might open in the core of the *c*-ring when the retracted *e* subunit pulls the L-PS plug out of the *c*-ring at the intracristae space, while the stabilized structure of ATP synthase pulls out phosphatidylserine at the matrix side. The mPTP opening dissipates the mitochondrial protonmotive force and water entries in the matrix driven by oncotic pressure.

**Figure 4 life-12-00401-f004:**
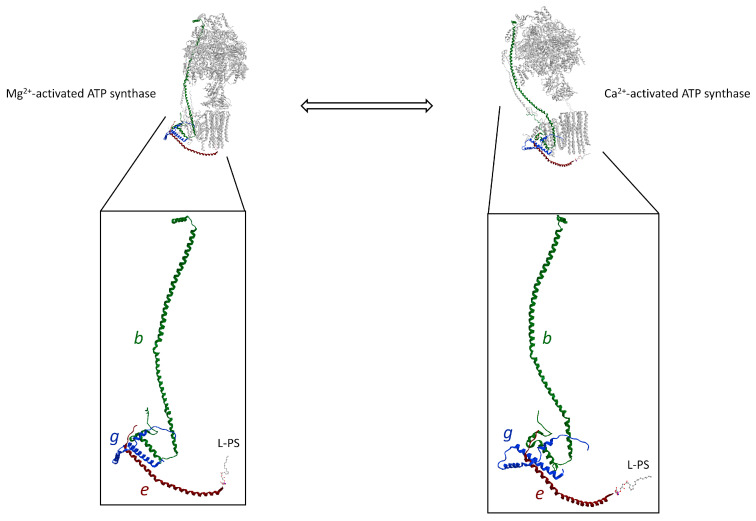
The “hook apparatus” of the ATP synthase (obtained from modified PDB ID code: 6TT7) with a TTMHB of *b*, *g*, and *e* subunits involved in pulling the lipid when the mPTP opens. The conformations of “hook apparatus” are depicted in the presence of the natural cofactor Mg^2+^ (**on the left**) and with Ca^2+^ (**on the right**). The differently coloured letters identify the subunits, drawn in the same colour as the letter. All other subunits of ATP synthase are shown in grey. The L-PS is illustrated as a ball-and-stick model.

**Table 1 life-12-00401-t001:** Clinical signs and symptoms of inherited ATP-synthase deficiency reported in single case report or patients’ cohort studies.

Gene	Variants	CNS	PNS	Muscle	Heart	EYE	Brain MRI	Metabolic Alterations	Other	References
**ATP5A1**	c.985C>T (p.Arg329Cys) c.962A>G (p.Tyr321Cys)	Encephalopathy Seizures Hypotonia	-	-	Pulmonary hypertension heart failure	-	Hyperintensity of thalamus and subcortical density Progressive cortical, subcortical, cerebellum and pons damage	Hyperalaninemia	Hypoplastic lungs renal cysts Microcephaly	[74,75]
**ATP5D**	c.245C>T (p.Pro82Leu) c.317T>G (p.Val106Gly)	Encephalopathy Lethargy Motor delay Ataxia Speech delay Myoclonic seizure Hypotonia	-	Proximal muscle weakness exercise intolerance	Dilatative cardiomyopathy	-	Transitory widespread cortical and subcortical oedema	Severe acidosis, Hypoglicaemia, Hyperammonaemia, 3-MGA^1^ Ketoacidosis	Short stature	[76]
**ATP5E**	c.35A>G (p.Tyr12Cys)	Ataxia	Peripheral neuropathy	Weakness exercise intolerance	Mild hypertrophy of left ventricle	-	-	Hyperlacticaemia, 3MGA^1^, hyperammonaemia	Respiratory distress	[78]
**DAPIT-UMSG5**	c.87+1G>C (p.?)	Motor delay Ataxia, Ophthalmoplegia	-	-	NA	-	Brainstem and basal ganglia lesions	-	NA	[79]
**MT-ATP6**	m.9176T>C (p.Leu217Pro) m.9185T>C (p.Leu220Pro) m.9127_9128delAT (p.Ile201ProfsX2) m.8993T>C (p.Leu156Pro) m.9185T>C (p.Leu220Pro) m.8993T>G (p.Leu156Arg) m.8993T>C (p.Leu156Pro) m.9017T>C (p.Ile164Thr) and m.9010G>A (p.Ala162Thr) m.9025G>A (p.Gly167Ser) m.9020A>G (p.His168Arg) m.9032T>C (p.Leu169Pro) m.9157G>A (p.Ala211Thr) m.8618insT (p.Ile31IlefsX64) m.8993T>G (p.Leu156Arg) m.8993T>G (p.Leu156Arg)	Cerebellar ataxia Myoclonic seizure Developmental delay Cognitive deficit Motor delay Seizures Hypotonia Spastic paraplegy Dystonia Dysarthria	Paraesthesia Motor and sensorineural neuropathy Perypheral polyneuropathy	Muscle weakness Muscle atrophy	Hypertrophic cardiomyopathy Supraventricular arrhythmia Atrioventricular block Bradycardia	Optic atrophy Retinal degeneration Retinitis pigmentosa Nystagmus, Strabismus, Ptosis Cataract Atypical LHON Blindness	Basal ganglia lesions Pituitary atrophy Medulla, pons and brain stem lesions White matter abnormalities Cerebellar vermis atrophy Cortical atrophy Cerebellar atrophy	Increase CSF lactic acid Lactic acidosis Increase of plasma pyruvate and alanine Metabolic acidosis Hyperammonaemia Increase of creatinine kinase	Sensorineural hearing loss Diabetes mellitus Hypogonadism HypothyroidismSurrenalic failure Short stature Respiratory distress Dysmorphisms Cafè-au-lait spot Deafness Headache Renal failure	[69,71,87,88,89,90,91,92,93,94,95,96]
**MT-ATP8**	m.8411A>G (p.Met16Val) m.8393C>T (Pro136Ser)	Developmental delay Ataxia Seizure Lethargy Encephalopathy Tetraplegia	Neuropathy	Weakness	Hypertrophic cardiomyopathy arrhythmia	Blindness	Cerebellar atrophy, White matter alterations	Lactic acidosis Hyperammonaemia Hyperalaninemia Hypoglicaemia	Deafness Anorexia Respiratory distress	[70,72]
**MT-ATP6** **/MT-ATP8**	m.8528T>C m.8529G>A m.8527A>G	Developmental delay Seizures	Global and axial hypotony	-	-	-	Lenticular nucleus and white matter anomalies	-	-	[73]
**ATPAF2**	c.280T>A (p.Trp94Arg)	NA	-	-	-	-	Cerebral atrophy Corpus callosum dysgenesis Hypomyelination Basal ganglia and thalamus atrophy	Urinary and plasma lactic acidosis	Dysmorphisms Hepatomegaly Renal hypoplasia	[80]
**TMEM70**	c.317-2A>G (p.?); c.628A>C (p.Thr210Pro) c.118_119insGT(p.Ser40CysfsTer11) c.317-2A>G (p.?) c.317-2A>G (p.?) Exon2 deletion	Encephalopathy Seizures Developmental delay Motor delay Ataxia Hypotonia	-	-	Dilated cardiomyopathy Arrhythmia Non-compact cardiomyopathy Hypertrophic cardiomyopathy	Cataract Ptosis	Cortical atrophy	3-MGA ^1^ Lactic acidosis, Hyperalaninemia hyperammonaemia Increase of ornithine	Dysmorphisms Microcephaly Hepatomegaly Umbilical hernia	[81,82,83,84,85,86,97,98]

^1^ 3-MGA: 3-metylglutaconic aciduria.

## Data Availability

The study did not report any data.

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
