# Peer review of "From the Structural and (Dys)Function of ATP Synthase to Deficiency in Age-Related Diseases"

_life, 2022, doi:10.3390/life12030401_

Round 1
Reviewer 1 Report
I appreciate the authors for compiling the review entitled “From the structural and (dys)function of ATP synthase to deficiency in age-related diseases” overall I find the review informative and interesting read. I only have these minor suggestions for improvement of the manuscript.
General comments:
- Language requires improvement throughout the review. Several sentences are very long and unclear.
- The role of ATP synthase complex in mitochondrial morphology is less investigated yet important aspect of mitochondrial form and function, the authors should cite more evidence supporting the notion. Additionally, recent reports has connected quality control of mitochondrial genome with Cristae morphology (10.1126/sciadv.abi8886), such reports should be discusses in the review for making it more innovative.
- Cyclophilin D is another critical factor for ATP synthase assembly, and its association with Alzheimer’s disease is relevant to the proposed review, therefore authors should compile one paragraph suggesting the role of cyclophilin D in ATP synthase assembly and disease.
- Abstract: I find the abstract unstructured and needs essential improvement. The authors moust consider restructuring of the abstract. It will be best to start with introducing the complex V and its importance in mitochondrial structure/function and energetics. Then factors affecting stability and function of the complex and their association with pathological conditions. At the end including the idea behind writing the review should be included.
Reviewer 2 Report
This well-written manuscript is a comprehensive review, which focuses on the molecular aspects of the mitochondrial ATP synthase complex with respect to its genetically determined deficiencies and neurodegenerative disorders. The topic of the review is clinically important and the relevant findings of the literature are discussed in a logical order with sufficient numbers of references. However, a few points should be checked before the final publication of the manuscript. Therefore, I recommend this manuscript for publication with minor modifications.
Specific suggestions:
- The labelling of amino acids (full name vs. three-letter codes) should be unified throughout the manuscript.
- All abbreviations should be given when first mentioned in the text.
- There are minor grammatical or spelling errors in lanes 29, 55, 79, 124, 150, 155, 158, 176, 220, 292, 313, 329, 359, 379, 380, 399, 400, and 420.
Author Response
This well-written manuscript is a comprehensive review, which focuses on the molecular aspects of the mitochondrial ATP synthase complex with respect to its genetically determined deficiencies and neurodegenerative disorders. The topic of the review is clinically important and the relevant findings of the literature are discussed in a logical order with sufficient numbers of references. However, a few points should be checked before the final publication of the manuscript. Therefore, I recommend this manuscript for publication with minor modifications.
We are pleased that the Reviewer considers our work of value and we would like to thank the reviewer for pointing out our errors. We have checked all these points and we have corrected them.
The labelling of amino acids (full name vs. three-letter codes) should be unified throughout the manuscript
Done, thanks.
All abbreviations should be given when first mentioned in the text.
Done, thanks.
There are minor grammatical or spelling errors in lanes 29, 55, 79, 124, 150, 155, 158, 176, 220, 292, 313, 329, 359, 379, 380, 399, 400, and 420.
Done, thanks.

Reviewer 3 Report
I believe that the presented article is a well-written review on a relevant topic. The authors made a thorough analysis of scientific literature. I think that the article doesn't require correction and could be published as presented.
Small note: Line 280 - "Charcot-Marie-tooth" (the last word must be capitalized)
Author Response
I believe that the presented article is a well-written review on a relevant topic. The authors made a thorough analysis of scientific literature. I think that the article doesn't require correction and could be published as presented.
We thank the Reviewer for agreeing that this review is useful to the field.
Small note: Line 280 - "Charcot-Marie-tooth" (the last word must be capitalized)
Done, thanks.
